

# An agroecological approach to preparation and use of a milk protein production baseline

Graeme D. Coles[1] and Jacqueline S. Rowarth[2]

[1] Ferrier Institute, Victoria University of Wellington, Wellington, New Zealand
[2] Agriculture and Life Sciences, Lincoln University Wellington, Lincoln, New Zealand

## ABSTRACT

Commercial dairy production occurs in a complex management environment, but increasingly, the dairy manager is expected to provide detailed reporting of productivity and environmental outcomes, for which conventional research methods double-blind crossover or case:control trials are inappropriate. This paper demonstrates the development of a milk protein production monitoring tool using a temporal (baseline) control in longitudinal, census-type investigations of modulation of system performance in response to factor change. It utilises farm-derived current and historical data, and contrasts seasonal responses with those achieved on neighbouring farms in a $2 \times 2$ contingency table. The approach is then shown to be useful in assessing the effect of two approaches to moderating milk urea concentration. Firstly, milk urea content can be monitored as it falls due to reduced feed protein content, and this fall can be arrested when milk protein content starts to decline relative to the value expected for the herd at any lactation stage. Secondly, by providing a dietary intervention aimed at increasing the availability of metabolic energy in the last month before calving, udder development can be augmented, leading to greater protein secretion capacity, meaning greater utilisation of circulating amino acids, and thus more limited substrate for urea synthesis. Thus, the changing impact of differing nutrition practices on dairy herd nitrogen excretion to environment can be followed with daily precision. In principle this approach can provide useful insights into a wide range of practical management interventions.

Corresponding author
Graeme D. Coles,
graeme.coles@richtech.co.nz

## INTRODUCTION

The Paris Agreement (*United Nations, 2015*) makes it plain that while initiatives to reduce the environmental impact of meeting global nutritional needs are needed, they should not compromise food production. *Coles, Porter & Wratten (2016)* showed that dairy production from grazed pasture is at least three times as effective in terms of land use as the best plant-based food system, when provision of essential amino acids (the first-limiting nutrient in adult diets) is considered. Dairy production is also a relatively efficient user of water (*e.g. Brown, Moot & Pollock, 2005*) compared with wheat production, and recently, has been shown to be a minor contributor to groundwater nitrate content against the background of other sources (*Rowarth & Coles, 2024*).
The origin of the amino acids for synthesising the desirable milk proteins that are the basis for the efficient provision of human nutrient needs is feed protein. Unlike monogastric animals, ruminants have ruminal microbial action that can convert dietary nitrogen, even from such unpromising sources as poultry excreta, to proteins with amino acid composition well suited to the metabolic needs of the animal (*Flachowsky, 1997*). However, even the most nutritionally desirable amino acids will be metabolised for energy, with release of their amino groups, if they are available in excess of the animal's ability to utilise them for maintenance, secretion or accumulation in body depots (*Schwab & Broderick, 2017*).

The amino acids surplus to dietary requirements used by livestock for metabolic energy production, have the nitrogen contained in the protein amino groups converted to urea and excreted (*Huntington & Archibeque, 2000*). A considerable proportion of this urea is captured by pasture plants during grazing, but while a small part of the unabsorbed urea nitrogen may reach ground water, the majority is lost to atmosphere (*Haynes & Williams, 1993*; *Fraser, 1994*; *Williams & Haynes, 1994*; *Laubach et al., 2013*; *Chadwick et al., 2018*; *Voglmeier et al., 2018*) where it contributes to greenhouse gas impacts in various ways (*Rubasinghege et al., 2011*; *Doane, 2017*).

The environmental impact of ruminant production (in the present case dairy production) has received considerable attention and in the New Zealand context, models, particularly "Overseer", have been developed in an attempt to predict the impact of ruminant production practices on environments outside the farm boundary. The opportunities and limitations of this approach have been intensively reviewed by the Parliamentary Commissioner for the Environment (*Upton, 2018*), and he has expressed reservations about the applicability of a modelling approach to impact determination, at least for regulatory purposes. Despite this, farmers face penalties for "wasting" nitrogen in the diet (*e.g.*, Fonterra's Purchased Nitrogen Surplus component of the Co-operative difference payment: https://www.fonterra.com/content/dam/fonterra-public-website/fonterra-new-zealand/campaign-images/codof/docs/2024-the-co-operative-difference-environment-factsheet.pdf).

Experimental evidence suggests that in the lactating dairy cow, minimisation of excess amino consumption can be monitored by determining the level of urea in milk as dietary protein availability (including digestibility) is decreased (*Lapierre & Lobley, 2001*; *Vibart et al., 2009*; *Pacheco et al., 2023*). However, there is a limit beneath which dietary protein supply should not fall (see below) if reduction in milk protein yield and body condition recovery is to be avoided. In this paper, we describe a set of methods by which a herd baseline milk protein production profile can be developed against which milk protein production in response to one or more dietary interventions can be compared. Milk urea concentration variation, as a rapid response to changes in amino acid catabolism, and an adequate predictor of urine urea daily excretion, may also vary seasonally, so baseline profiles for this parameter should also be prepared.

A "temporal (baseline) control in longitudinal, census-type investigations of system performance in response to factor change" is proposed. This is more than the 'paired farm' approach that has been used in comparisons of farm systems *e.g.*, organic and conventional (*Reganold, 2013*), fertiliser advice (*Bryant et al., 2019*), or catchments (*Schilling, Jones &*

*Seeman, 2013*) because the factors changed in the baseline approach are identified and outcomes can be linked directly to drivers.

A number of advantages suggest themselves. Firstly, provided the herd in question is managed on a single site, maintaining a consistent age structure, environmental impacts on herd performance will be restricted to climatic variation and known management interventions. These can be accounted for in ANOVA if required, but in any case, are likely to bear similarly on all herd members, even if individuals respond idiosyncratically. Secondly, farmer interest is in whole-herd performance, achieved in a moderately well-known management system which is maintained in a consistent way from season to season, with only incremental changes, on the basis that "if it ain't broke, don't fix it". It is not for nothing that assumptions drawn from the myriad of sampling-based human investigations conducted in each quinquennium are tested against data from the censuses taken every five years. Thus, we propose that sound conclusions can be drawn from comparisons of whole-herd current season's data in response to one or more management changes relative to a baseline constructed from an adequate body of historical data. Conclusions drawn can be further strengthened if adequate "replication" over farms using similar management and genetics is tested by a 2 × 2 contingency approach. In this paper, we test this approach and show how it can be used, both to evaluate proposed management changes (including the use of new products), and report the outcomes of such changes as required.

Dairy production, particularly in pasture-based systems, with calving over a concentrated period in either spring or autumn, or both, is influenced by climate, and management responses thereto. Consequently, comparison of performance due to a management change cannot simply be contrasted to a single previous lactation. Instead, it is necessary to provide a baseline of performance based on mean performance over a number of lactations, which, if not consistent with one another, are at least well-characterised in their variation. Standard approaches to Analysis of Variance (ANOVA and ANCOVA) are suitable tools for this purpose.

A key consideration for satisfactory analysis is that the time-course results for each season are properly temporally aligned. Simply using calendar date to align different lactations may confound analysis, because physiologically-important events occur in response to seasonal factors such as droughts that may necessitate temporary dietary changes, for instance.

It is known that for an individual cow, peak milk volume production occurs the day she conceives (*Olori et al., 1997*). Not every cow in a dairy herd will conceive on the same day, but there will be a day on which the largest number of cows are mated, and this will lead to an apparent peak in milk yield.

From the point of conception, the cow commences a process of gestation, drawing resources away from milk production towards embryo growth. Inevitability, milk production will fall to a minimum prior to delivery of a subsequent calf, and initiation of colostrum synthesis for the neonate. The shape of the production decline curve is consistent among individual cows (Fig. 1) but the shape of the rise to peak production curve depends, in part, on when conception occurs.

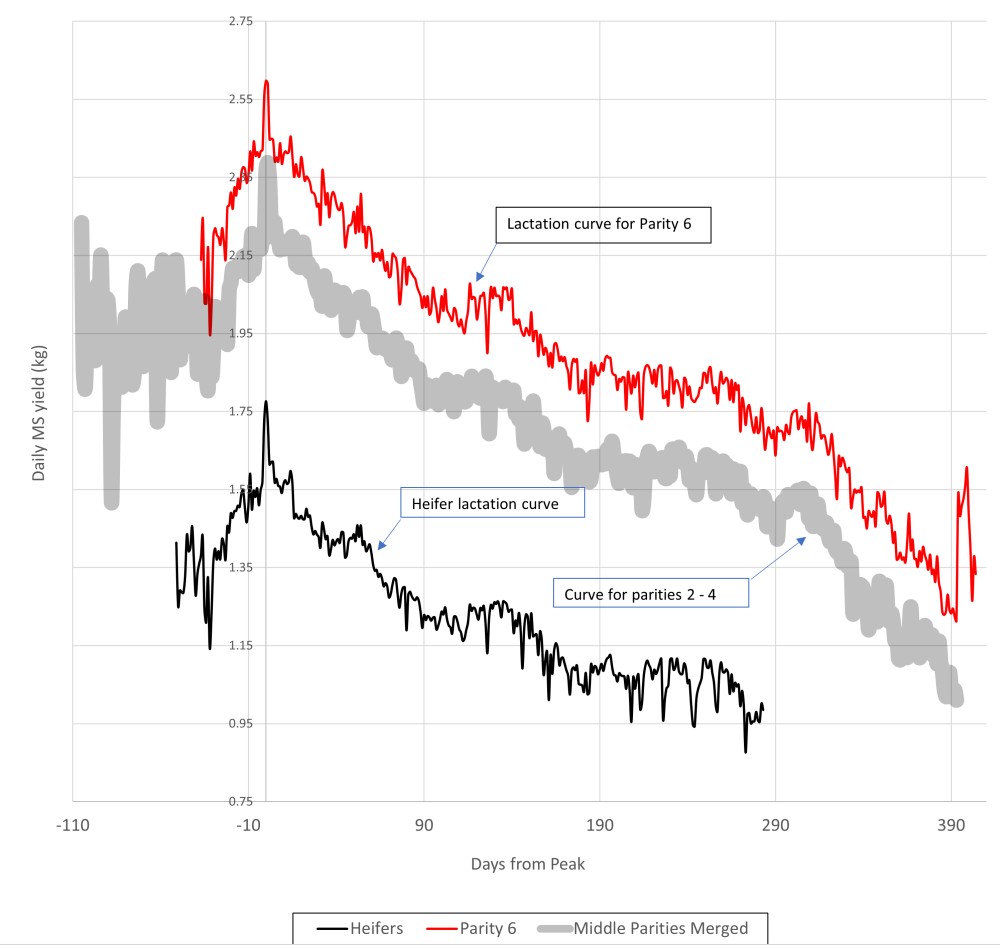

**Figure 1   Milk solids yield curves for robotically-milked Jersey cows, ranging in parity from 1 to 6.**
Note that in the study from which this analysis was drawn, data for were lacking for parity 5. Since results
for parities 2 to 4 varied only randomly, they were merged to produce the broadened, partly transparent
curve labelled ''Curve for Parities 2 –4'' See Tables 1 and 2 for statistical information.

This paper discusses an approach to examining nitrogen use within a farmer-managed
herd in response to a change in inputs, in comparison with a similar farmer-managed herd
without the change in inputs, and compared with previous years. The aim is to provide a
robust approach to measuring effects under non-ideal research conditions. It also provides
a real-time measurement alternative to modelling when attempting to estimate nutrient
losses to the larger environment from specific farming operations.

## METHODS

### Animal ethics

All data used in the preparation of this paper were acquired from databases of information
previously collected by farmers for commercial management purposes, and therefore, were
not obtained by direct animal intervention. Under New Zealand Animal Welfare standards,
prior animal ethics approval is not required to use such data (K. Littin, Manager, National

Animal Welfare Advisory Committee, Ministry for Primary Industry 2022, *pers. comm.*).
Knewe®-Mg is a chelate of magnesium with 3- and 4-carbon organic acids with prebiotic
effects (*Coles & Pearce, 2011*) that has been assessed by the New Zealand Agricultural
Chemicals and Veterinary Medicines Board, and found to be exempt from classification
requirement, so can be freely fed to all classes of livestock.

## Standard lactation curve preparation

The information presented in Fig. 1 and Tables 1 and 2 was developed from data acquired
from a housed, robotically-milked mixed-age herd of Jersey cows derived from New Zealand
blood lines. Cows may have calved on any day, and be mated when deemed appropriate
by the herd manager. Cows may have been dried off before their next parturition in the
period covering from two months to one week before expected calving.

From this herd, animals following a normal annual lactation length (albeit that they
calved at any time of the year), were selected and their milk and milk solids production
patterns analysed. Data from approximately three years were processed.

Raw data were derived from daily herd-wide records presented in a proprietary .html
format, and required significant processing to prepare them for statistical analysis. Steps in
this processing were:

1. Conversion of proprietary-format (Time for Cows, Lely Industries N.V. Maassluis, The
   Netherlands) information to a form where individual lactations could be selected and
   checked for compliance with the criteria described above. In practice, it was necessary
   to use Quattro® Pro X9 (Perfect Office, Corel Corporation) for this purpose, as it was
   the only utility capable of carrying out the interpretation of the proprietary .html data
   format to produce a format suitable for further analysis;
2. Concatenation of monthly data files to allow detection of individual lactation milk
   flows;
3. From the full set of production profiles those terminating within 365 days of calving
   were selected for further analysis. A total of 30 sets of production data were found that
   complied with this criterion. Data were smoothed using a five-point moving average,
   and aligned relative to day of peak production. Animals fell into six parity classes, but
   no data for parity 5 were obtained, since all animals in this parity were milked for
   longer than 365 days.
4. Data were submitted to Minitab® v20 for analysis of variance. A restricted maximum
   likelihood model was used to account for the fact that data were markedly unbalanced.
   Day from peak production and animal were treated as random variables, while parity
   number was a fixed variable.

**Table 1** REML analysis of variance results for lactation curves analysis.

| Variance components Source | | | | | |
|---|---|---|---|---|---|
| Source | Var | % of Total | SE Var | 95% CI | Z-Value |
| Days from peak | 0.096162 | 47.64% | 0.006830 | (0.0836651, 0.110525) | 14.079157 |
| Animal Number | 0.052035 | 25.78% | 0.014758 | (0.0298448, 0.090723) | 3.525751 |
| Days from peak*Parity | 0.003576 | 1.77% | 0.000666 | (0.0024822, 0.005153) | 5.367138 |
| Error | 0.050068 | 24.81% | 0.000790 | (0.0485429, 0.051641) | 63.355286 |
| Total | 0.201841 | | | | |
| Source | P-Value | | | | |
| Days from peak | 0.000 | | | | |
| Animal Number | 0.000 | | | | |
| Days from peak*Parity | 0.000 | | | | |
| Error | 0.000 | | | | |
| Total | | | | | |

$-2$ Log likelihood = 710.900656.

| Tests of fixed effects | | | | |
|---|---|---|---|---|
| Term | DF Num | DF Den | F-Value | P-Value | Kenward & Roger λ |
| Parity | 4.00 | 25.05 | 5.59 | 0.002 | 1.00000 |

| Model summary | | | | |
|---|---|---|---|---|
| S | R-sq | R-sq(adj) | AICc | BIC |
| 0.223759 | 73.77% | 73.76% | 718.90 | 747.99 |

| Coefficients | | | | | | |
|---|---|---|---|---|---|---|
| Term | Coef | SE Coef | DF | 95% CI | T-Value | P-Value |
| Constant | 1.639284 | 0.050961 | 29.25 | (1.53510, 1.74347) | 32.167639 | 0.000 |
| Parity | | | | | | |
| 1 | −0.504434 | 0.113347 | 25.04 | (−0.73785, −0.27101) | −4.450359 | 0.000 |
| 2 | 0.023736 | 0.067044 | 25.08 | (−0.11432, 0.16179) | 0.354040 | 0.726 |
| 3 | 0.109916 | 0.101171 | 25.04 | (−0.09843, 0.31826) | 1.086436 | 0.288 |
| 4 | 0.078244 | 0.101171 | 25.04 | (−0.13010, 0.28659) | 0.773386 | 0.447 |

**Notes.**

From this analysis, a set of fitted values for each parity were obtained and are presented as lactation curves in Fig. 1.

Note that in this analysis, the data are significantly skewed due to the outlying behaviour of the parity 1 (heifer) individuals (Table 2).

## Baseline development

Fonterra is New Zealand's largest milk processor, acquiring approximately 85% of the country's annual production from approximately 10,000 suppliers (http://www.Fonterra.com accessed 26th February 2022). Milk is collected each day (or each second day at the beginning and end of lactation), the collected volume estimated, and samples taken for analysis. Analytes include:

- Milk fat concentration;
- Milk protein concentration;
- Milk urea concentration;

**Table 2 Grouping parity information using the Tukey method and 95% confidence interval.**

| Parity | N | Mean | Grouping |
|---|---|---|---|
| 6 | 1,507 | 1.93182 | A |
| 3 | 1,411 | 1.74920 | A |
| 4 | 1,389 | 1.71753 | A |
| 2 | 5,362 | 1.66302 | A |
| 1 | 973 | 1.13485 | B |

**Notes.**
  Means that do not share a letter are significantly different.

- Somatic cell count, and;
- Fat Evaluation Index.

From these are calculated:

- Total Milk Solids;
- Total Fat and;
- Total protein.

Milk solids yield per cow and per hectare are also presented. Each milk supplier has access to their own data by logging into their account at https://nzfarmsource.co.nz/. For the purposes of this investigation, the authors were given login rights to each farm account. For the purposes of this study, data for seven farms from the three most recent seasons (2018–9, 2019–20 and 2020–21) were downloaded as .csv files.

## Data alignment

The first step in preparing data for analysis is to check for internal consistency of each data set. Daily milk volume estimates were checked in case volumes from more than one collection on a single day were combined, leading to over-estimation of herd production on that day, and if this is the case, volume allocated appropriately. At the beginning and end of the season, milk collections are made each second day. In these cases, the collected milk volume is split evenly between the collection day and the previous day.

Data were transferred to a statistics package (Minitab® 20.4) and subjected to ANOVA using the Generalised Linear Model. Factors in the model used were:

- Year and Days from peak milk flow (yield data were either individual daily values or smoothed values using a 5-point moving average)
- Year and Days from index date (1st July in each season). In this case, only individual values were analysed.

In all three analyses, both Year and Day were treated as fixed variables. An initial analysis was carried out, and aberrant values determined by reference to residual plots, primarily the normal probability plot of residuals and the plot of residuals against observation order. Aberrant values detected in this way were excluded from the data set, and the GLM ANOVA repeated. Not more than 10% of data were excluded in any analysis. For more information on GLM formulas, see the following website: https:

**Table 3  Coefficients of variation (%) for each approach to data alignment for all seven farms' data for milk yield.**

| Herd | Date alignment | Days from peak | Smoothed days from peak |
|---|---|---|---|
| Tatua | 5.80% | 4.68% | 5.75% |
| Nikau | 4.75% | 6.04% | 4.00% |
| Rimu | 5.11% | 6.43% | 6.52% |
| Miromiro | 3.86% | 4.91% | 3.79% |
| Ruamanu | 8.32% | 9.76% | 9.54% |
| Kimberley | 7.20% | 10.82% | 10.82% |
| Ring | 3.20% | 8.35% | 8.03% |
| Mean | 5.46% | 7.28% | 6.92% |

//support.minitab.com/en-us/minitab/help-and-how-to/statistical-modeling/anova/how-to/fit-general-linear-model/before-you-start/example/.

The analyses were conducted in such a way as to provide fitted means for daily volume by year and days from the index day. The ANOVA provided both a fitted mean and an SE value for each factor. To assess the most appropriate method for aligning data, the mean of all daily SE values was calculated and divided by the overall trial mean, thus providing a coefficient of variation, expressed as a percentage (Table 3).

It is evident from this analysis that the smallest amount of unexplained variance is achieved using date alignment.

It should be noted that the number of cows milked each day varied from time to time. This was particularly evident at the start of lactation, as cows calved and entered the milking herd. Cows were also withdrawn from production temporarily if, for instance, they suffered a case of mastitis, or were medicated for any other reason. It is therefore important to ensure that the number of cows contributing to each collection is accurately known.

## RESULTS

Once data had been checked for internal consistency, and if necessary, daily data from different seasons smoothed and aligned, a milk volume baseline curve was prepared (Fig. 2).

Figure 3 shows the comparison between the baseline curve and the milk yield curve for the same herd in the 2021–22 year, during which it was fed Knewe®-Mg (Knewe Biosystems NZ Ltd, Omihi Road, Greta Valley, NZ) according to the manufacturer's specification, commencing 4 weeks before the expected mean calving date for the herd. By comparing the total volume collected for the period of interest in 2021 (orange spots) with the baseline estimate for the same period (day 25 to day 126), the impact of the treatment could be estimated (a yield increase of 8.1%).

Fitted means for milk yield were determined using days from index date and days from peak, and regressed on one another. $R^2$ for the relationship was 0.9926 ($p = 0.00$) and the intercept was not different from 0 ($p = 0.059$), meaning that the two approaches to data alignment gave results not different to one another. Therefore, the following results were

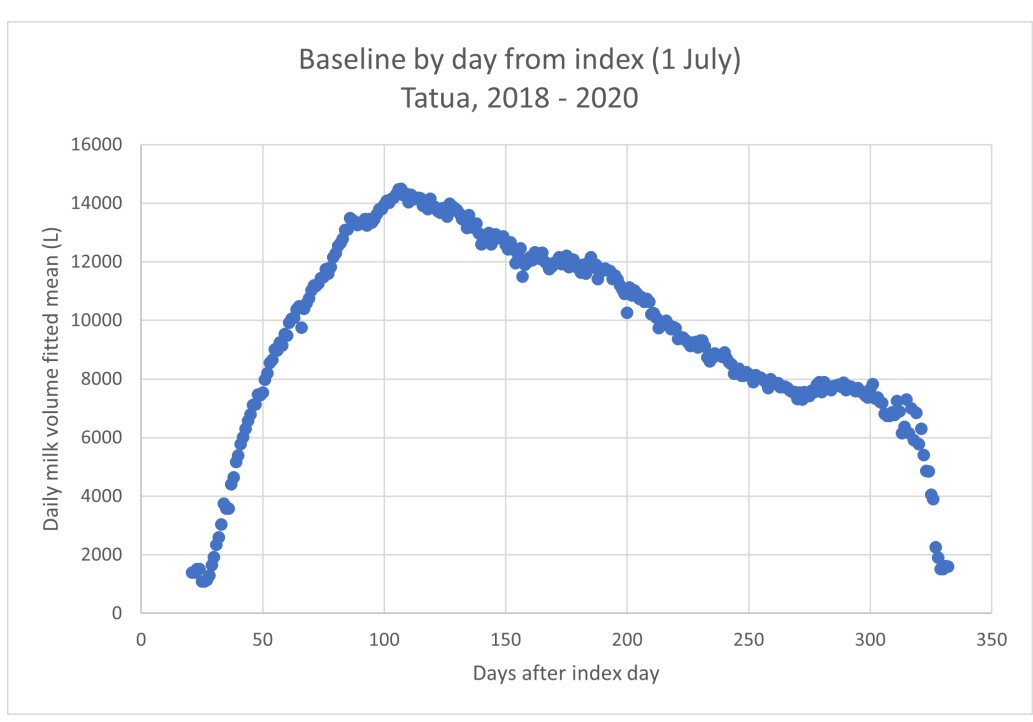

**Figure 2** **Baseline milk volume curve for a single farm.** The baseline is estimated based on milk yields at days after July 1st. Breed is Friesian.

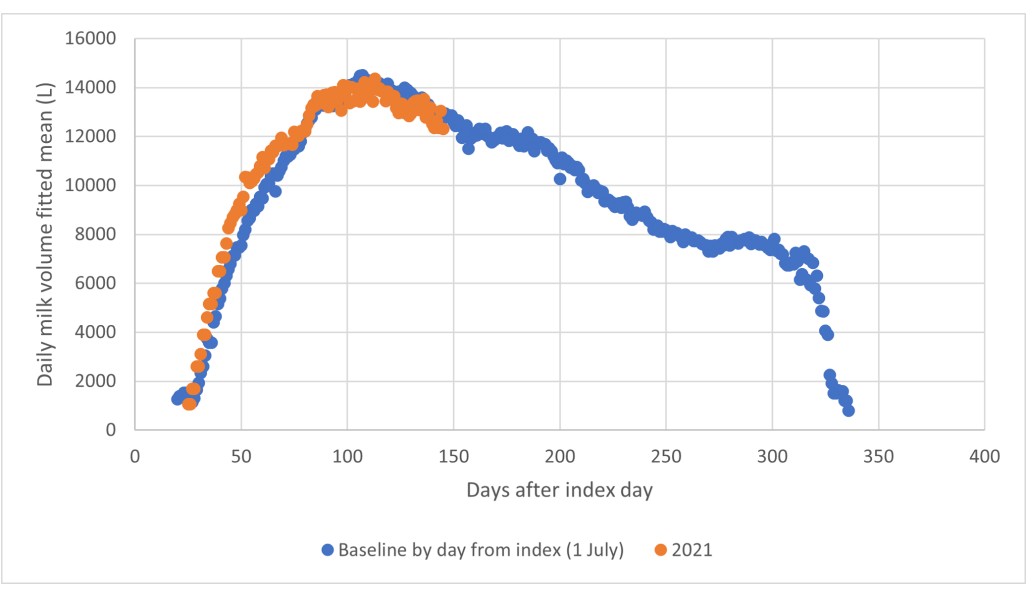

**Figure 3** **Comparison of daily milk volume (litres collected) for the commencement of the 2021–22 dairy season with the baseline derived from the previous three seasons.** The yield advantage over this period is 8.1%.

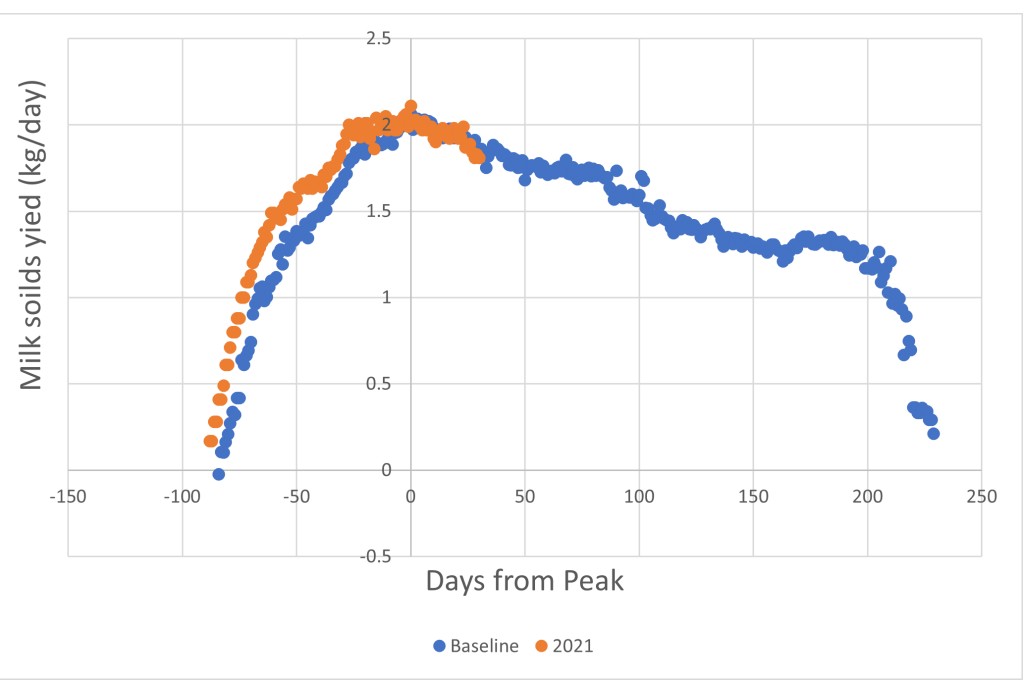

**Figure 4** Comparison of daily milk solids produced for the commencement of the 2021–22 dairy season with the baseline derived from the previous three seasons. Per cow yield advantage was 7.1%.

prepared using days from peak volume as the basis for data alignment. The reason for this will become evident below.

## Milk composition baseline determination

Using the information provided from Fonterra's Farmsource database, a similar baseline for daily milk solids yield **per cow** was constructed and compared with the values for the start of the 2021 season (Fig. 4).

Similarly, a pair of curves for milk protein yield was prepared (Fig. 5).

Milk urea concentration is known to be well correlated with daily urine urea excretion, since both depend on the level of serum urea in the cow (*Gustafsson & Palmquist, 1993*; *Pacheco et al., 2023*). Figure 6 shows **herd** milk urea concentration levels.

It is notable that over the same period that milk protein yield was **elevated** by 7.5%, milk urea concentration was **depressed** by 16.2%. It is also notable that milk urea concentration showed regular fluctuations that coincided in each of the three years, when data were aligned according to days from peak milk production. This coincidence of periodic fluctuation was not evident if date alignment is used.

## Farm-to-farm comparison

Neighbouring farms enjoy similar climatic conditions during production seasons, and when in unified ownership, are expected to be subject to similar management. It is reasonable, therefore, to contrast performance relative to baseline between a farm and a neighbour managed by the same entity. The farm whose baselines were calculated above

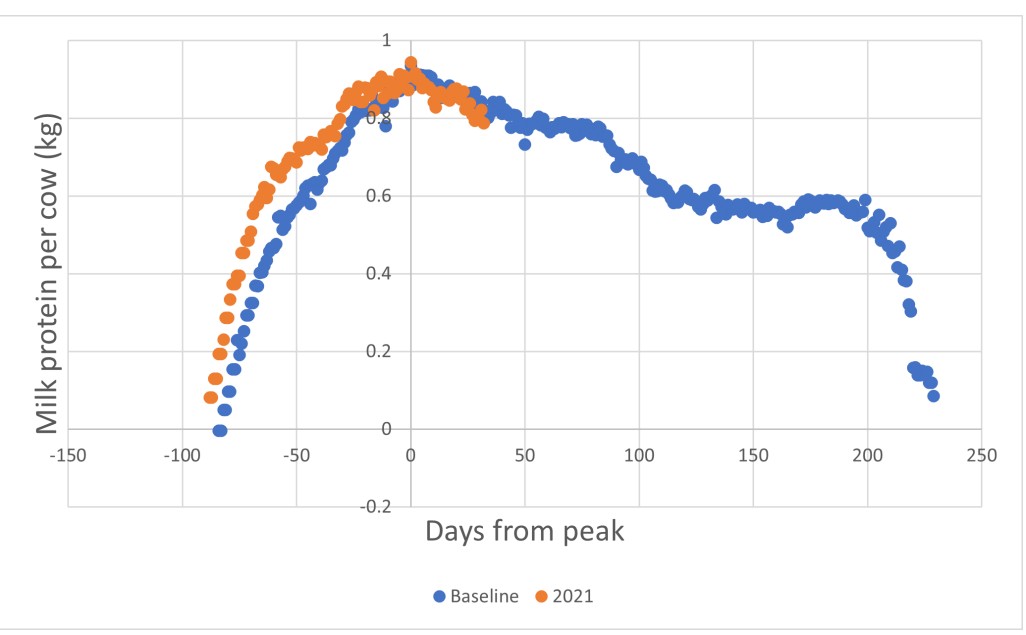

**Figure 5** **Milk protein yield curves for Tatua Farm.** The per-cow advantage achieved over the period covered was 7.5%.

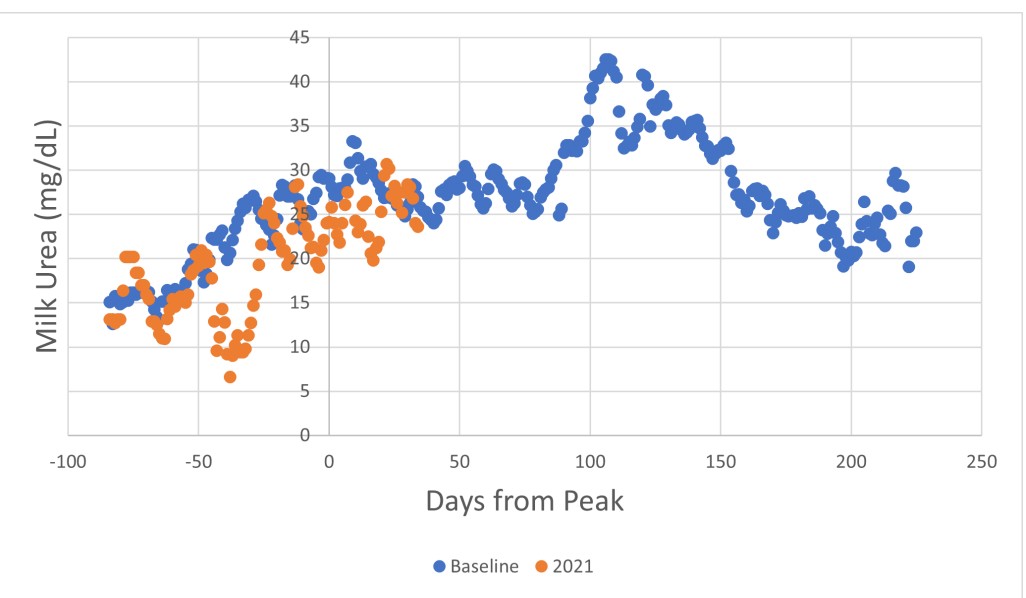

**Figure 6** **Daily milk urea concentration (mg/dL) for Tatua farm.**

has a neighbour (Nikau) with a herd of animals of similar genetic background, and access to similar pasture and bought-in forage. Daily milk yield is shown in Fig. 7.

It will be noted that the daily fitted values were significantly higher than for Tatua. This is due to the higher number of cows in the Nikau herd, so a comparison was made of the

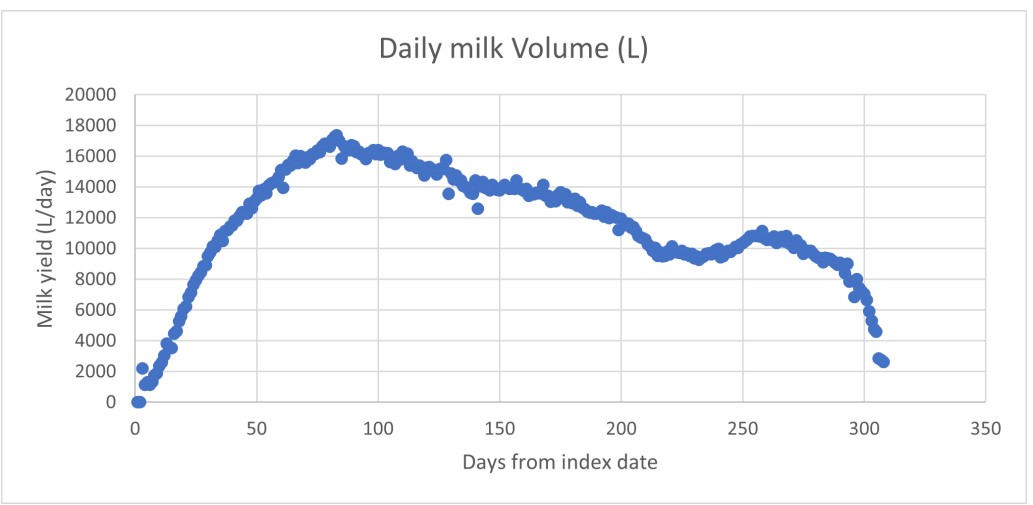

**Figure 7** Daily milk volume fitted values for Nikau farm.

**Table 4** Relative mean annual daily per cow milk solids production (kg cow$^{-1}$ day$^{-1}$).

| Season | Tatua | Nikau | Ratio |
|---|---|---|---|
| 2018–19 | 1.321 | 1.779 | 1.347 |
| 2019–20 | 1.506 | 1.750 | 1.162 |
| 2020–21 | 1.453 | 1.749 | 1.204 |
| Overall mean | 1.427 | 1.759 | 1.233 |

**Notes.**
All differences $p = 0.00$.

daily milk solids production **per cow** in the two herds across the three years. Results are shown in Table 4 and Fig. 8.

Once the baseline performance of the two farms had been established, it was possible to relate performance changes due to specific interventions with at least some confidence. Table 5 provides an example.

Despite the looseness of control occasioned by the fact that the two farms are neighbours and may have been subjected to differing management across the three years data from which the performance baselines are determined, the scale of the impact of the management intervention is quite obvious. A similar comparison of daily milk protein yield was instructive (Table 6).

The daily yield of milk protein is relatively fixed compared with milk fat (*National Research Council, 1988*), in response to management intervention modifying metabolic energy availability. The assumption drawn from these observations is that milk protein synthesis is limited by the capacity of the udder, and that there is excess substrate in the form of necessary amino acids in well-nourished cows. In one case, where Knewe®-Mg was added to a conventional supplemented pasture diet, milk urea level doubled to ∼80 mg/dL, then fell back to 30 mg/dL when 400 g of crude protein was removed from the diet, with no change in milk protein secretion. The increase in milk protein seen in the feeding
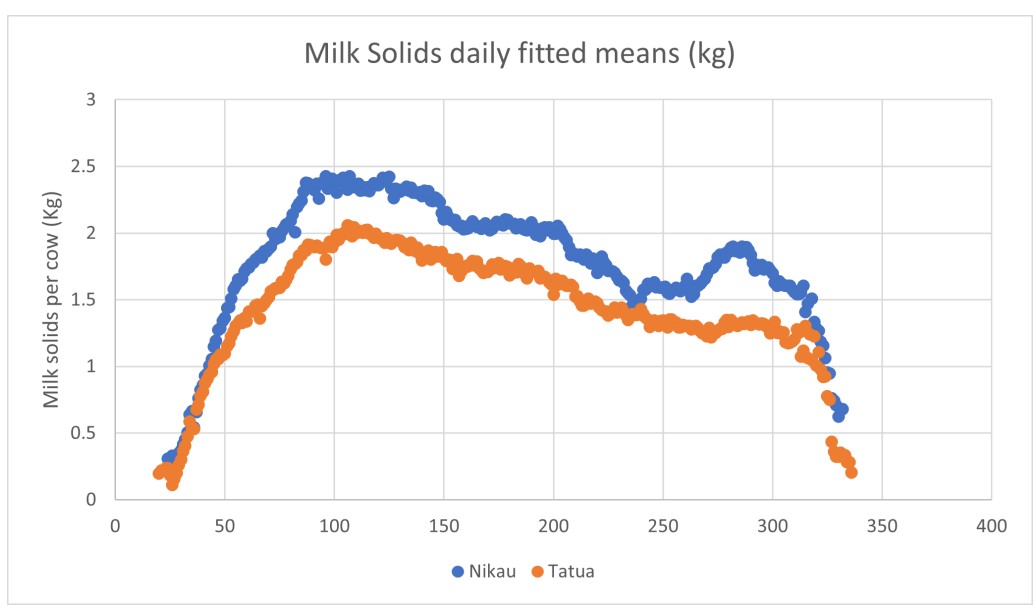

**Figure 8** **Comparison of daily milk solids yields per cow for two neighbouring farms.** Regression analysis shows that the two curves are very similar, with only a scaling difference ($R^2 = 0.948$, $p = 0.00$).

**Table 5** Impact of feeding Knewe®-Mg on per-cow daily milk solids production.

| Farm | Baseline (kg/day) | 2021–22 season(kg/day) | Difference |
|---|---|---|---|
| Tatua (+Knewe®-Mg) | 1.46 | 1.624 | +11.23% |
| Nikau (-Knewe®-Mg) | 1.851 | 1.456 | −21.3% |

study reported here was therefore markedly surprising. The data shown in Fig. 6 above were derived from the same feeding study, and together with the information provided in Table 4 show that it is possible to lift protein secretion capacity (presumably through an increase in udder size during springing (*McClary, Rapnicki & Overton, 2014*) and that this can be accompanied by a substantial reduction in nitrogen excretion.

The nett reduction in protein intake in the example given above was 320 g per day, and this led to a reduction of milk urea concentration of ∼50 mg/dL, with no loss of protein secretion. Virtually all the surplus nitrogen is either recycled through forage for grazing or lost to atmosphere as $NH_3$, and since the cycle is repeated approximately 12 times in a season, >95% of the original surplus nitrogen consumed is converted to $N_2O$, then ultimately to $N_2$ (*Crutzen, 1979*)

From the above the surplus protein converted to $N_2O$ is 300 g/day, which equates to

$$((300/6.25) \times 44/28 \text{ g}) = 75.4 \text{ g}$$

of $NO_x$ ultimately reaching the atmosphere daily and since $NO_x$ is a long-lived gas, the annual $GHG_e$ saved by reduction of urea excretion is

**Table 6  Comparison of management effects on daily per cow milk protein yield.**

| Farm | Baseline (kg/day) | 2021–22 season (kg/day) | Difference |
|---|---|---|---|
| Tatua (+Knewe®-Mg) | 0.672 | 0.721 | +10.7% |
| Nikau (-Knewe®-Mg) | 0.875 | 0.646 | −26.2% |

$(75.4 \times 0.3 \times 298 \times 365/1,000) = 2.460$ tonnes of $CO_2$ equivalent,

or 45% more than the impact of the gross methane emissions of the same cow.

## DISCUSSION

As described above, considerable investment has been made in attempts to model nutrient losses from individual farms based on relatively sparse trial results (*Upton, 2018*). The results of such modelling are being used for regulatory purposes, but there is considerable scepticism as to their reliability at the individual farm scale.

Numerous attempts have been made to produce reliable mathematical models of lactation curves similar to those presented in Fig. 1 (*Koonawootrittriron et al., 2001*; *Khazaei & Nikosiar, 2005*; *Silvestre, Petim-Batista & Colaco, 2006*; *Cankaya, Unalan & Soydan, 2011*; *Macciotta et al., 2011*; *Vinay-vadillo et al., 2012*; *Jeretina, Babnik & Škorjanc, 2013*; *Correa-Luna et al., 2020*). However, the degree of fit of mathematical approximations to observed curves is not particularly high, and fails to take into account the possibility of transient events that regularly occur in practical farming situations. In pasture-based management these may include transient drought, adverse weather events, parasite outbreaks, and even changes in farm staff. Given that detailed reliable performance information is available, it seemed logical to use it to make season-to-season and farm-to-farm comparisons.

Relative to the types and range of variations in animal performance within a dairy herd, and the consequent difficulties accompanying any effort to select truly representative samples of animals from such herds, the "agroecological" approach, in which current-season performance of the whole herd, compared with a long-run average performance, has obvious attractions. Adopting an agroecological approach (in the scientific sense of the word: *Wezel et al., 2009*; *Wezel et al., 2020*), allows a reliable and repeatable estimation of the extent of a change induced by a novel management intervention in the operation of a farm system, in this case a dairy herd. It is noted that the concept of agroecology has been progressively co-opted towards something governing not only the interaction of agricultural practice with its immediate environment, but also food production systems and their contribution to smallholder welfare (*Wezel et al., 2020*). This co-option not only misuses a concept of considerable current importance, but leads to possible misinterpretation of historic reporting of the results of investigations in the context of specific agricultural practice. Because we have found particular value in this approach, we wish to use the term "agroecology" in the sense of its traditional meaning.

What are the alternatives? The "gold standard" for determining the nature and extent of a benefit conferred by a product or management change in a biological system is the double-blind randomised crossover experimental design (*Schulz, Altman & Moher, 2010*) but for many such products or management changes, this design is inappropriate, for practical reasons. This design assumes that the underlying physiology of the subject varies only randomly in the course of the trial, but for treatment arms requiring months to years for putative effect to appear (see Fig. 1) individuals such as lactating cows cannot act as their own control.

Case-control investigations, in which an attempt is made to ensure that investigation arms are populated by matched individuals, are the next best alternative. However, even this approach suffers from a number of possible difficulties. Firstly, there is a risk that sampling of the population in which the investigation is to be conducted will not take into account any lack of statistical normality. Once again, Fig. 1 is informative. Every dairy herd will include a certain proportion of 1st parity animals, but these animals are clearly different to adults, in that they conceive well before they reach their adult body weight, and continue to grow during their first lactation. What is more, it is likely that animals failing to conceive in time to calve within twelve months of their previous parturition will be concentrated in this age cohort. Given that the average number of lactations achieved by a New Zealand dairy cow is 5.17 (2020–21 data from DairyNZ (*LIC & Dairy NZ, 2021*)) this first parity cohort will comprise about 20% of dairy herds, providing a major cause of systematic variation.

The relatively small coefficients of variation achieved in forming baseline lactation curves from three seasons' data suggest that the approach we have developed provides reliable and repeatable benchmarks against which to compare current season's results to assess the impact of management interventions. As shown in Table 1, great statistical detail is available to assess the extent of any change due to such interventions, while Table 2 shows that the effect of an intervention can be presented in a form that is easily comprehensible in the farm management situation. Information derived in this way can be reported to any desired level of detail to meet regulatory needs, and to drive further change.

## CONCLUSIONS

The approach suggested here to describing and measuring on-farm changes achieved in response to interventions of practical value can provide insight not otherwise available, particularly using modelled approaches. Coefficients of variation estimated by this approach are small, indicating relatively high precision of estimation of treatment responses. From a practical standpoint, impacts of management changes can be observed in a timely manner, allowing flexible, prompt responses. Furthermore, farmer reporting to third parties can be carried out on a consistent basis, allowing easy comparisons to be made between enterprises operating in similar environmental frameworks.

## ACKNOWLEDGEMENTS

We are most grateful to Andrew Fletcher of Fonterra, for supporting access to the data on which our approach has been tested, and for detailed discussions of approaches to data alignment. Similarly, we are grateful to van Leeuwen Group, Mark Bridges of Southern Pastures and Eliot Cooper of Cooper Deleste for generous access to their data. Paul Tocker, Max Enersen and Annette Marr provided technical support and tolerated flights of fancy until cygnets turned into swans.

### Funding

No external funding was received for the preparation of this work.

### Competing Interests

Graeme Coles is a director of Knewe Biosystems Ltd, the beneficial owner of Rich Technology Solutions Ltd. Jacqueline Rowarth is a farmer-elected director of Dairy NZ and of Ravendown Ltd. None of these entities have had any input into this submission.

### Author Contributions

- Graeme D. Coles conceived and designed the experiments, performed the experiments, analyzed the data, prepared figures and/or tables, authored or reviewed drafts of the article, and approved the final draft.
- Jacqueline S. Rowarth analyzed the data, authored or reviewed drafts of the article, and approved the final draft.

### Animal Ethics

The following information was supplied relating to ethical approvals (i.e., approving body and any reference numbers):

All data used in the preparation of this paper were acquired from databases of information previously collected by farmers for commercial management purposes, and therefore, were not obtained by direct animal intervention. Under New Zealand Animal Welfare standards, prior animal ethics approval is not required to use such data (K.Littin, Manager, National Animal Welfare Advisory Committee, Ministry for Primary Industry 2022, pers. comm.). Knewe®-Mg is a commercial product that has been assessed by the New Zealand Agricultural Chemicals and Veterinary Medicines Board, and found to be exempt from classification requirement, so can be freely fed to all classes of livestock.

### Data Availability

The data is available in the Supplemental Files.

### Supplemental Information

Supplemental information for this article can be found online at http://dx.doi.org/10.7717/peerj.18103#supplemental-information.

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
