# Peer review of "An agroecological approach to preparation and use of a milk protein production baseline"

_PeerJ, doi:10.7717/peerj.18103_

## Round 0.1 · original submission · Major Revisions

Dear Dr. Coles,

If you feel you can revise your manuscript according to the reviewers' comments, please revise your manuscript and submit it. Please also send us your written responses to each of the reviewers' comments.

Yours,

Yoshi

Prof. Yoshinori Marunaka, M.D., Ph.D.

Reviewer 1 ·

Basic reporting

The present article provides useful information about an agroecological approach to preparation and use of a milk protein production baseline. It is in general appropriately organized, carried out and written, however there are some points that should be corrected.
Abstract: Rationale/context of the study and findings are missing
Introduction: some of literature needs updated/old publications. Merge sections (Challenges in developing a baseline for a biological system and Seasonal influences) with introduction part focussing on specific gaps in knowledge you wish to address, and the importance/relevance of your work
Discussion is missing or shallow next to your findings. Please provide the essential interpretation based on key findings; compare and contrast with previous/recent studies on other approaches.

Experimental design

No comment

Validity of the findings

No comment

Reviewer 2 ·

Basic reporting

General comments
In the manuscript, you need to harmonize the use of "approach" and "method" to avoid any confusion. Explain which principles of agroecology are addressed by the developed approach. Refer to the article by Wezel et al. (2020) for inspiration and highlight how the proposed approach is agroecological and which agroecological principles it refers to.
Here is the article’s reference: Wezel A, Gemmill Herren B, Bezner Kerr R, Barrios E, Rodrigues Gonçalves A L, Sinclair F, 2020. Agroecological principles and elements and their implications for transitioning to sustainable food systems. A review. Agronomy for Sustainable Development (2020) 40: 40. https://doi.org/10.1007/s13593-020-00646-z
Overall, the manuscript needs to be reviewed to clearly distinguish between the results and the discussion. In the latter, the proposed approach should be compared with other approaches to ensure what it brings that is new, and to identify its advantages and disadvantages.

Abstract
Line 19: Specify the two approaches used in parentheses.
Lengthen the abstract further by highlighting the main significant results of the tests conducted for the two approaches.
Insert between 3 and 5 keywords at the end of the abstract.
Introduction
Line 35 : You should write the year of publication of the article instead of using "op.cit".

Challenges in developing a baseline for a biological system
Lines 60-63 : In these sentences, “Once again, Figure 1 is 61 informative. Every dairy herd will include a certain proportion of 1st parity animals, but these animals are 62 clearly different to adults, in that they can conceive well before they reach their adult body weight, and 63 continue to grow during their first lactation”, delete “Once again, Figure 1 is 61 informative” and insert at the end of the sentence after first lactation ‘’figure 1”.

Tables
The tables as currently presented do not effectively highlight the obtained results and need to be reorganized. Thus, Table 1 should be presented succinctly, including the structuring with the grouping done in Table 2. Tables 1 and 2 should be combined, highlighting the parameters, the mean, the SE, the structuring of the means with letters a, b, as well as the probability.
For Table 3, instead of presenting the coefficients of variation, it is necessary to present the means, SE, and probabilities to compare the milk yields of the seven farms, as these are quantitative variables.
Tables 5 and 6 should be combined to present not only the impact of feeding Knewe®-Mg on per-cow daily milk solids production but also the comparison of management effects on daily per-cow milk protein yield.

Experimental design

No comment

Validity of the findings

In-depth work still needs to be done to highlight the novelty of the approach, as well as its advantages and disadvantages.

Reviewer 3 ·

Basic reporting

Improve English, update old references

Experimental design

All information on the work was understandable and consistent with each other.

Validity of the findings

However, the author (s) left something to be desired in the discussion of the results. They could have explored the discussion better, emphasizing the importance of this study, for example.

Additional comments

Work more discussions

Reviewer 4 ·

Basic reporting

The paper need an improvement in presentation for a clearer understanding of the readers. Major areas required improvements are indicated in the manuscript.

Experimental design

The research was well designed and executed by the presentation of the methods and approach requires an improvement which has been indicated in the corrected manuscript. The authors are advised to make the corrections for replication of the study by other researchers.

Validity of the findings

The findings are valid.

Additional comments

Here is a general comment to provided as constructive feedback on improving the paper's structure:

While the content of the paper is valuable, the overall structure needs improvement to enhance readability and ensure a clearer presentation of findings. To achieve this, the authors should consider the following suggestions:

1. Ensure the introduction provides a comprehensive overview of the study's background, significance, and objectives. Clearly state the research question and the study's aims.

2. Incorporate a detailed literature review paragraph that contextualizes the research within existing studies. This will help readers understand the relevance and novelty of your work.

3. The authors need to present the methodology in a structured and detailed manner. Clearly describe the study design, data collection methods, and analytical techniques. This will allow readers to follow your research process, assess its validity, and potential for replication.

4. The results have to be organized logically, with clear headings and subheadings. Present your findings systematically, citing your tables, graphs, and figures to enhance clarity and visual appeal.

5. Pay attention to the overall formatting and flow of the paper. Use clear and consistent headings, subheadings, and transitions between sections. Ensure that paragraphs are well-structured and concise.

By addressing these points, the structure of the paper will be significantly improved, making it easier for readers to follow your research and understand the significance of your findings.

Annotated reviews are not available for download in order to protect the identity of reviewers who chose to remain anonymous.

---

## Round 0.2 · accepted · Accept

DDear Dr. Coles,

I am very pleased to accept your manuscript. In accepting your manuscript, I have only one request.

As reviewer 2 pointed out, please insert the equation of the model used for the GLM in lines 182-183 during the proofreading stage.

Yours,

Yoshi

Prof. Yoshinori Marunaka, M.D., Ph.D.

Reviewer 2 ·

Basic reporting

No comment

Experimental design

No comment

Validity of the findings

Lines 182-183: Insert the equation of the model used for the GLM, as this will allow for a better assessment of the validity of the results.

Reviewer 4 ·

Basic reporting

The manuscript has been improved. All sections of the paper has been improved and it makes it clearer and easier to understand. The results were presented in a clearer and better manner for easier and quicker understanding.

Experimental design

No comment.

Validity of the findings

No comment.

Additional comments

No comment.